# Website Premia for Extensive Margins of International Firm Activities: Evidence for SMEs from 34 Countries †

Joachim Wagner

Institute of Economics, Leuphana Universität Lüneburg, D-21335 Luneburg, Germany; wagner@leuphana.de
† The data from the Flash Euromarometer 421 can be downloaded free of charge after registration at http://www.gesis/eurobarometer. Stata code used to generate the empirical results reported in this note is available from the author.

**Abstract:** This paper uses firm-level data from the Flash Eurobarometer 421 survey conducted in June 2015 in 34 European countries to investigate the link between having a website and international firm activities in small- and medium-sized enterprises (SMEs). We find that firms that are present on the web more often export, import, engage in research and development cooperation with international partners, work as subcontractors for firms from other countries, use firms in other countries as subcontractors, and perform foreign direct investments—both inside and outside the European Union. The estimated website premia are statistically highly significant after controlling for firm size, country, and sector of economic activity. Furthermore, the size of these premia can be considered to be large. Internationally active firms tend to have a website.

**Keywords:** website premia; international firm activities; Flash Eurobarometer 421

## 1. Motivation

Presence on the web is today considered as an important part of a firm's strategy to successfully make a living. This tends to be even more important in times of the COVID-19 pandemic, when quarantines and lockdowns increase the costs of face-to-face contact with (potential) buyers and sellers. Wagner (2021) uses firm-level data from the World Bank Enterprise surveys conducted in 2019 and from the COVID-19 follow-up surveys conducted in 2020 in ten European countries to investigate the link between having a website before the pandemic and firm survival until 2020. The estimated positive effect of web presence is statistically highly significant ceteris paribus after controlling for various firm characteristics that are known to be related to firm survival. Furthermore, the size of this estimated effect can be considered to be large on average. Similarly, Muzi et al. (2022) report, based on firm-level data collected for 34 economies up to 18 months into the COVID-19 crisis, that businesses that have a website are more likely to continue existing. A website helps firms to survive.

Furthermore, Wagner (2022) shows that, in 2019, firms from 18 European countries that had a website were larger, older, more productive, and more often exporters, product innovators, process innovators, and (partly) foreign-owned firms than firms without a website. Good firms tend to have a website.

Given this high importance of a web presence for the performance of firms, it comes as a surprise that there seems to be no comprehensive evidence on the existence of websites in firms with various forms of international activities.[1] This note contributes to the literature by reporting descriptive evidence on the share of firms with a website in 34 European countries in 2015 based on the Flash Eurobarometer 421 surveys conducted among small- and medium-sized enterprises in these countries. We look at differences between firms with

and without a website in the use of various types of international activities using estimates of so-called website premia that show the percentage difference between firms with and without a website, controlling for firm size, country of origin, and sector of economic activity of the firm.

We expect these website premia to be positive for international activities for two reasons.

First, having a website reduces information costs for potential business partners in other countries. Potential importers and customers in other countries can more easily see details of the products or services provided. Producers in foreign countries can more easily see which products or services a firm in another country that is a potential importer of their products or services might be interested in. A match of partners in cooperative agreements on research and development projects is more easily initiated, and the same holds for agreements in subcontracting (as principal or agent).

Second, from results reported in Wagner (2022) for firms from 18 European countries (based on data from different surveys conducted in 2019), we know that older, more productive, and innovative firms tend to have a website—and these firm characteristics are known to be positively related to international firm activities.

To anticipate the most important result, we find that firms that are present on the web are more active internationally—they more often export, import, engage in research and development cooperation with international partners, work as subcontractors for firms from other countries, use firms in other countries as subcontractors, and perform foreign direct investments—both inside and outside the European Union. The estimated website premia are statistically highly significant after controlling for firm size, country of origin, and sector of economic activity. Furthermore, the size of these premia can be considered to be large. The take-home message, therefore, is that internationally active firms tend to have a website.

The rest of the paper is organized as follows. Section 2 introduces the data used and discusses the international firm activities that are looked at. Section 3 reports results from the econometric investigation. Section 4 concludes.

## 2. Data and Discussion of Variables

The firm-level data used in this study are taken from the Flash Eurobarometer 421 survey conducted in June 2015 in 34 European countries.[2] All firms are small- and medium-sized enterprises (SMEs) with 1 to 249 employees.

In the survey, firms were asked in question Q11_1, "Is it possible to look at a website presenting your products and/or services?" Firms that answered "yes" are classified as firms with a web presence.

Descriptive evidence on the share of firms with a web presence in the total sample and by country is reported in Table 1. While the overall share of firms with a website in the sample is 75.75 percent, figures differ widely between the 34 countries. Web presence is only 49.00 percent in Albania and 49.64 percent in Bulgaria, while 88.98 percent of all firms in the sample have a website in Sweden and 92.59 percent in Denmark.

At the bottom of Table 1, the share of firms with a website is reported by sector of (main) economic activity of the firm. While firms from manufacturing and services are more often present on the web compared to the overall average figure, and firms from retail and industry have a lower rate of web presence, the figures do not differ by order of magnitude.

In the empirical investigation of the link between web presence and various types of international firm activities, firm size is controlled for. Firm size is measured as the number of employees (in full-time equivalents) at the time of the survey; see question D1a.

In the empirical study, we look at various types of international firm activities inside and beyond the European Union.

We consider exports, imports, working with a partner based abroad for research and development (R&D) purposes, working as a subcontractor for a company based abroad, using a subcontractor for a company based abroad, and investing in a company based

abroad. If a firm stated in the survey that it did one of these activities in the last three years (in question q2 with regard to countries inside the European Union or in question q3 with regard to countries outside the European Union), it is considered as an exporter to the EU, etc. Each firm, therefore, can be active or not in 12 different types of international activities.

In questions Q6 and Q10, firms were asked to which countries they exported in 2014, or which countries they did import from, respectively. The answers were coded in a way that allowed the computation of the number of destinations exported to or imported from between zero and nine (see the questionnaire for details).

Furthermore, firms are divided by broad sectors of activity (manufacturing, retail, services, and industry) based on information in variable nace_b.

Descriptive statistics for all variables are reported for the whole sample used in the empirical investigation in Appendix A, Table A1.

**Table 1.** Share of SMEs with web presence, 2015.

| Country/Sector | Number of Firms | Share of Firms with Website (Percent) |
|---|---|---|
| All | 13,710 | 75.75 |
| Albania | 100 | 49.00 |
| Austria | 476 | 87.61 |
| Belgium | 487 | 82.14 |
| Bulgaria | 421 | 49.64 |
| Croatia | 481 | 79.21 |
| Cyprus | 193 | 67.88 |
| Czech Republic | 474 | 86.71 |
| Denmark | 486 | 92.59 |
| Estonia | 466 | 65.45 |
| Finland | 486 | 85.39 |
| France | 468 | 70.09 |
| Germany | 467 | 83.51 |
| Greece | 486 | 81.07 |
| Hungary | 468 | 78.21 |
| Iceland | 184 | 76.09 |
| Ireland | 471 | 76.43 |
| Italy | 466 | 63.30 |
| Latvia | 479 | 69.52 |
| Lithuania | 480 | 67.08 |
| Luxemburg | 194 | 72.68 |
| Makedonia | 169 | 51.48 |
| Malta | 197 | 72.08 |
| Moldavia | 191 | 62.30 |
| Montenegro | 194 | 77.32 |
| Netherlands | 468 | 85.47 |
| Poland | 444 | 73.87 |
| Portugal | 479 | 67.85 |
| Romania | 480 | 55.42 |
| Slovak Republic | 495 | 82.83 |
| Spain | 486 | 75.10 |
| Sweden | 490 | 88.98 |
| Turkey | 473 | 82.03 |
| United Kingdom | 440 | 80.45 |
| Manufacturing (NACE C) | 2982 | 80.05 |
| Retail (NACE G) | 4216 | 73.53 |
| Services (NACE H,I,J,K,L,M,N) | 4007 | 78.36 |
| Industry (NACE B,D,E,F) | 2505 | 70.22 |

Source: Own calculation based on Flash Eurobarometer 421; see text for details.

### 3. Testing for Website Premia in International Firm Activities

　　To test for the difference in the types of international firm activities listed in Section 2 between firms with and without a website, and to document the size of these differences, an empirical approach is applied that modifies a standard approach used in hundreds of empirical investigations on the differences between exporters and non-exporters that has been introduced by Bernard and Jensen (1995, 1999). Studies of this type use data for firms to compute so-called exporter premia, defined as the ceteris paribus percentage difference in a firm characteristic—e.g., labor productivity—between exporters and non-exporters. These premia are computed from a regression of log labor productivity on the current export status dummy and a set of control variables:

$$\ln LP_i = a + \text{ß Export}_i + c \text{ Control}_i + e_t \tag{1}$$

where i is the index of the firm, LP is labor productivity, Export is a dummy variable for current export status (1 if the firm exports, 0 else), Control is a vector of control variables, and e is an error term. The exporter premium, computed from the estimated coefficient ß as $100(\exp(\text{ß}) - 1)$, shows the average percentage difference between exporters and non-exporters controlling for the characteristics included in the vector Control (see Wagner (2007) for a more complete exposition of this method).

　　Here, we look at differences between firms with and without a website (instead of differences between exporters and non-exporters) and are interested in the existence and size of website premia instead of exporter premia (see Wagner 2022). For international firm activities that are measured by dummy variables (the 12 extensive margins listed in Section 2), the empirical model is estimated by Probit instead. Therefore, (1) becomes (2)

$$\text{Indicator}_i = a + \text{ß Website}_i + c \text{ Control}_i + e_{it} \tag{2}$$

where i is the index of the firm, Indicator is a dummy variable for the use of a form of international firm activity, Website is a dummy variable for the presence of a website in the firm (1 if the firm has a website, 0 else), Control is a vector of control variables (that consists of a measure of firm size, and dummy variables for countries and sectors of economic activity), and e is an error term. The website premium is computed as the estimated average marginal effects of the website dummy variable.

　　For the number of markets in exports or imports, (1) becomes (3)

$$\ln \text{number}_i = a + \text{ß Website}_i + c \text{ Control}_i + e_{it} \tag{3}$$

where i is the index of the firm, number is the log of the number of markets in exports or imports, Website is a dummy variable for the presence of a website in the firm (1 if the firm has a website, 0 else), Control is a vector of control variables (that consists of a measure of firm size, and dummy variables for countries and sectors of economic activity), and e is an error term. Note that due to the log transformation, only firms that export to or import from at least one destination are included in the computations.

　　The website premium, computed from the estimated coefficient ß as $100(\exp(\text{ß}) - 1)$, shows the average percentage difference between firms with and without a website, controlling for firm size, country of origin of the firm, and the broad economic sector that it is active in.

　　Results are reported in Table 2. The broader picture that is shown is perfectly clear: firms that are present on the web are more often active in international economic activities. The estimated website premia are statistically highly significant ceteris paribus after controlling for firm size, country, and sector of economic activity. Furthermore, the size of these premia can be considered to be large.

**Table 2.** Website premia (percent) for margins of international firm activities.

| Variable | Premia | Prob-Value |
|---|---|---|
| Export to EU countries | 14.35 | 0.000 |
| Import from EU counties | 16.41 | 0.000 |
| R&D cooperation within EU | 4.92 | 0.000 |
| Subcontractor (agent) within EU | 4.58 | 0.000 |
| Subcontractor (principal) within EU | 6.92 | 0.000 |
| Foreign direct investor within EU | 0.48 | 0.000 |
| Export to Non-EU countries | 0.31 | 0.000 |
| Import from Non-EU countries | 11.52 | 0.000 |
| R&D cooperation beyond EU | 3.23 | 0.000 |
| Subcontractor (agent) beyond EU | 3.75 | 0.000 |
| Subcontractor (principal) beyond EU | 4.15 | 0.000 |
| Foreign direct investor beyond EU | 1.51 | 0.000 |
| Number of export markets | 20.17 | 0.000 |
| Number of import markets | 10.25 | 0.000 |

Source: Own calculations with data from Flash Eurobarometer 421. The website premium shows the average percentage difference between firms with and without a website, controlling for firm size, country of origin of the firm, and the broad economic sector it is active in; for details, see text.

However, it is an open question (that is asked in the same way when exporter premia are discussed) whether these premia are due to the self-selection of more internationally active firms into web presence or whether these premia are the effect of having a website.

## 4. Concluding Remarks

This paper demonstrates that having a website is positively related to international firm activities. Website premia are large for all types of international firm activities looked at here. Does this study imply that in order to be active in international markets, firms should have a website, or that having a website will help the firms to be internationally active? This is an open question (that is asked in the same way when exporter premia are discussed) because we do not know whether these premia are due to the self-selection of internationally active firms into web presence, or whether they are the effect of having a website. This cannot be investigated with the data at hand. To answer this important question, longitudinal data for firms are needed that cover several years and that include a sufficiently large number of firms that switch their status between having a website and not over time (in both directions). To the best of my knowledge, such data are not available as of today. Let us collect it!

**Funding:** This research received no external funding.

**Informed Consent Statement:** Not applicable.

**Data Availability Statement:** Details are provided in the note to the title of the article, see page 1.

**Conflicts of Interest:** The author declares no conflict of interest.

## Appendix A

**Table A1.** Descriptive statistics for sample (N = 13,710) used in estimations.

| Variable | Mean | Std. Dev. |
|---|---|---|
| Web presence | 0.758 | 0.43 |
| (Dummy; 1 = yes) | | |
| Firm size | 31.88 | 44.02 |
| (Number of employees) | | |

**Table A1.** *Cont.*

| Variable | Mean | Std. Dev. |
|---|---|---|
| Export to EU countries | 0.395 | 0.49 |
| (Dummy; 1 = yes) | | |
| Import from EU countries | 0.492 | 0.50 |
| (Dummy; 1 = yes) | | |
| R&D cooperation within EU | 0.100 | 0.30 |
| (Dummy; 1 = yes) | | |
| Subcontractor (agent) within EU | 0.158 | 0.36 |
| (Dummy; 1 = yes) | | |
| Subcontractor (principal) within EU | 0.185 | 0.39 |
| (Dummy; 1 = yes) | | |
| Foreign direct investor within EU | 0.045 | 0.21 |
| (Dummy; 1 = yes) | | |
| Export to Non-EU countries | 0.262 | 0.44 |
| (Dummy; 1 = yes) | | |
| Import from Non-EU countries | 0.277 | 0.45 |
| (Dummy; 1 = yes) | | |
| R&D cooperation beyond EU | 0.051 | 0.22 |
| (Dummy; 1 = yes) | | |
| Subcontractor (agent) beyond EU | 0.072 | 0.26 |
| (Dummy; 1 = yes) | | |
| Subcontractor (principal) beyond EU | 0.091 | 0.29 |
| (Dummy; 1 = yes) | | |
| Foreign direct investor beyond EU | 0.030 | 0.17 |
| (Dummy; 1 = yes) | | |
| Number of export markets | 0.860 | 1.43 |
| Number of import markets | 0.867 | 1.17 |

Source: Own calculations with data from Flash Eurobarometer 421; for details, see text.

## Notes

[1] See Gopalan et al. (2022) for website presence and simultaneously exporting and importing, Huang and Song (2019) for internet use and exports in China, Li et al. (2022) for internet use and firms' exports and imports in China.

[2] See Table 1 for a list of the countries covered in the survey. Data are available free of charge after registration from http://www.gesis/eurobarometer (accessed on 1 September 2022).

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
