# Peer review of "Website Premia for Extensive Margins of International Firm Activities: Evidence for SMEs from 34 Countries†"

_economies, doi:10.3390/economies10100250_

Round 1

Reviewer 2 Report

The subject of the article is interesting. However, the author cites a secondary study from 2015 (Flash Eurobarometer 421 (Internationalization of Small and Medium-Sized Enterprises).

There has been a major shift in the website premium for extensive margins of international firms over the past seven years, especially during the pandemic and current geopolitical instability.

Thus, please complete:

1. Introduce conclusions to the tables concerning the figures from 2015.

2. Explain how the situation in the study area has changed in the last seven years. Determine the impact of the pandemic and inflation.

Round 2

Reviewer 1 Report

Your justifications are relevant.